# COVID-19 Lockdown Stress and the Mental Health of College Students: A Cross-Sectional Survey in China

**Ziao Hu, Jun Li ***, **Ling Pan and Xiaoying Zhang**

School of Finance and Economics, Hainan Vocational University of Science and Technology, Haikou 571126, China
\* Correspondence: lijun.edu.ma@foxmail.com

**Abstract:** To prevent the spread of the COVID-19 pandemic, countries around the world adopted varying degrees of lockdown. The lockdowns restricted the freedom of college students, which led to stress and mental health issues. This study constructed a mediating model to explore the relationship between COVID-19 lockdown stress and Chinese college students' mental health; the mediating role of fear of missing out (FoMO) was also investigated. A 7-item COVID-19 student stress questionnaire (CSSQ), a 6-item mental health scale, and a 10-item FoMO scale were distributed among 695 college students who experienced lockdown in China. The results showed that COVID-19 lockdown stress was significantly and negatively correlated with mental health, significantly and positively correlated with FoMO, and FoMO was significantly and negatively correlated with mental health. COVID-19 lockdown stress significantly and negatively influenced Chinese college students' mental health directly and indirectly via the complementary partial mediating effect of FoMO. The results intensify our comprehension of the influence of COVID-19 lockdown stress and mental health problems in Chinese college students and also provide practical suggestions for college educators to address such scenarios.

**Keywords:** COVID-19 lockdown stress; mental health; fear of missing out; basic psychological needs theory; Chinese college students

## 1. Introduction

The World Health Organization deemed COVID-19 a public health emergency that poses a severe threat to human health [1]. To prevent the development of COVID-19, the governments of China and other countries were forced to implement various containment measures [2–6], which led to dramatic changes in people's daily lives and led to passive and specific type of stress [2,4]. China closed all schools during this period and restricted all students to their homes, which restricted students' outdoor activities and prevented them from interacting with their peers [7]. The lockdown challenged the academic life patterns of college students as well as created a state of social isolation that had a particularly pronounced psychological impact on them [8]. A study noted that the prevalence of mental health problems among Chinese students increased from 17% to 28% during the pandemic [9].

Isolation and stress have been proven to be associated with various psychological problems [10]. Studies have shown that stress during COVID-19 lockdown is significantly and negatively associated with college students' mental health [11–13]. Another study including 283 college students revealed that stress negatively affected mental health through insomnia [14]. Gunnell et al. [15] indicated that COVID-19 lockdown stress is a cause of mental illness and even suicide.

FoMO, as a state of unmet basic psychological needs [16], fits deeply with the isolated social environment during the epidemic. China's strict lockdown policy restricted students' normal social activities during the epidemic [7], which may have posed a threat to the

satisfaction of college students' basic psychological needs, such as autonomy and interpersonal relationships. Previous studies have pointed out that the inability to meet basic psychological needs is the theoretical source of conceptualization of FoMO [17]. In addition, recent research results also showed that lockdown brings new stress to college students [2], and stress positively and significantly predicts FoMO [18]. A systematic literature review from 2002–2020 found that FoMO was extremely closely related to mental health [17] and was a crucial influencing factor for mental health [14]. During the COVID-19 lockdown, numerous studies on stress or psychological problems have discussed FoMO as a critical mediating variable [18–21]. Therefore, this study explored the mediating role of FoMO between COVID-19 lockdown stress and college students' mental health.

According to basic psychological needs theory (BPNT), environmental changes affect the satisfaction of basic psychological needs and behavioral motivation in humans [22,23]. The failure to satisfy primary psychological needs can lead to mental health problems in humans [24,25]. Although there is no dearth of prior research on stress on mental health issues, in a specific social context, COVID-19 lockdown stress is a new stress response [2] that requires renewed research attention. Moreover, recent research has not considered the combined effects of COVID-19 lockdown stress and behavioral motivation (FoMO) on college students' mental health. This study therefore attempts to construct a mediation model to explore these effects. At the same time, the mental health of college students during the epidemic had received continuous attention [12,13,26]. The results of this study will provide a practical reference for colleges and universities to guide college students to maintain a positive and sustainable mental health state during the COVID-19 lockdown.

## 2. Literature Review

### 2.1. Basic Psychological Needs Theory (BPNT)

BPNT, proposed by Deci and Ryan in the 1980s, is the most central subtheory of self-determination theory; BNPT argues that the environment influences the satisfaction of basic human psychological needs and behavioral motivation [22,23]. Three basic psychological needs essential to individual behavior and psychological well-being were proposed, namely, the need for autonomy (the need for individuals to have a sense of independence in the activities they engage in rather than being controlled by others), the need for relatedness (the need for individuals to relate to others or to belong to a group), and the need for competence (the needs for individuals to be able to perform certain activities, to express their abilities, and to have control over their environment) [27,28]. BPNT was considered in various studies and applied to mental health research during the pandemic; it was proven to have strong explanatory power and generalizability [24,25,29–31].

Research has shown that the eruption of the COVID-19 pandemic and associated containment policies have negatively affected mental health by posing significant challenges to these three basic human psychological needs [25]. Specifically, containment policies to prevent the spread of an epidemic can limit people's freedom in exercising their daily activities [2], which can reduce their decision-making autonomy in life processes [25]. Due to the social nature of humans, social relationships are one of the essential elements of human life [32]. However, the "isolation" introduced by lockdowns cuts people off from actual social activities and increases the social isolation of college students [33], leaving basic interpersonal needs unmet. In addition, lockdowns confine people to a limited physical space, which reduces individuals' connection with others and the outside world as well as limits individuals' abilities to explore the environment, creating a sense of insecurity from the uncontrollable situation of the pandemic and their lives [25,34]. These unmet basic needs due to lockdown have been shown to cause stress [2,29] as well as psychological distress, such as anxiety and maladjustment [29,30,35]. It also triggers FoMO among college students [36], which in turn poses a serious threat to their mental health [14,37,38]. Recent studies have pointed out that basic psychological needs theory was the source of conceptualization of FoMO, which was closely related to the generation of FoMO and mental health [17].

To sum up, this study believed that COVID-19 lockdown may cause college students' basic psychological needs to be unmet, thus generating FoMO and threatening mental health. Therefore, this study assumed that FoMO plays a mediating role between COVID-19 lockdown stress and college students' mental health.

## 2.2. COVID-19 Lockdown Stress and Mental Health

The Chinese government adopted strict control policies to prevent the widespread transmission of the COVID-19 epidemic [3,5,6]. In contemporary research, similar emergency public health events, such as SARS, have been shown to cause stress and significantly affect individual psychological health [4,39]. Extreme changes in daily life are a significant source of stress [4]. By contrast, a series of drastic changes in life and learning, such as school closures and restrictions on activity space during COVID-19, exert a negative psychological impact on the adolescent population, and it is particularly pronounced compared to other populations [8,40]. Research confirms that the lockdown or quarantine of the epidemic caused severe [12,41] and multidimensional psychological stress to university students [2]. For example, pandemics and blockades bring new stressors to medical undergraduates' studies [42]. A longitudinal survey by Hakami et al. [43] showed that the COVID-19 lockdown elevated stress among college students. Zurlo et al. [2] also showed that the COVID-19 lockdown led to stress among college students in three aspects, namely, interpersonal relationships and academic life, social isolation, and fear of contagion.

Isolation and stress during lockdown usually results in various psychological problems, such as depression, sleeplessness, stress, irritability, and anxiety [10]. Adams et al. [14] pointed out that stress can positively predict the degree of mental health impairment among college students. Studies have indicated that stress is significantly related to mental health during COVID-19 lockdowns [11–13]. Specifically, a longitudinal study by Savage et al. (2020) on university students in the UK showed that perceived stress was negatively correlated with mental health status, that university students' stress increased significantly, and that their mental health deteriorated considerably after five weeks of COVID-19 lockdown. Studies have also confirmed that higher stress among college students results in them experiencing more psychological problems [13]. A literature review on stress and mental health showed that traditional learning of college students was abruptly and forcibly switched online during the COVID-19 lockdown, removing them from conventional learning environments; their inability to communicate and interact with their peers and lecturers led to increased levels of stress (and anxiety), which negatively impacted their mental health [12].

Based on the above discussion, hypothesis 1 was proposed: COVID-19 lockdown stress is significantly and negatively correlated with and predicts mental health among college students.

## 2.3. Mediating Role of FoMO

FoMO is defined as "a pervasive concern that others may have beneficial experiences in one's absence" [16]. FoMO begins by generating feelings of missing out and then using compulsive behaviors to maintain social connections [17]. Studies have shown that social isolation during COVID-19 lockdown caused reduced personal socialization [44], and individuals with inadequate social needs were more prone to developing FoMO [45,46]. In addition, FoMO can further promote interpersonal needs [47], putting them in a vicious circle [26]. Recent studies have shown that stress is positively associated with FoMO among adolescents [18,48]. An investigation on Chinese university students confirmed that stress was significantly and positively correlated with, and also predicted, FoMO [18].

In addition, studies have affirmed that FoMO is significantly related to mental health [14,37,38]. Barry et al. (2017) studied psychosocial adjustment of US adolescents and showed that higher FoMO implied more prominent mental health problems. Luca et al. (2020) pointed out that FoMO was critical in increasing mental health problems (e.g. psychological mood and mental disorders). Adams et al. (2020) also confirmed that FoMO levels in college students significantly and negatively predicted mental health.

FoMO has often been discussed as an essential mediator in recent studies on stress or psychological problems [18–21]. For example, research confirmed that stress further promotes inappropriate mobile phone use among Chinese college students via FoMO [18]. Wang et al. [21] suggested that FoMO partially mediates the effects of stress on online gaming disorders. Ergin and Ozer [20] showed a significant partial mediating role of FoMO between social detachment and phobia. Some studies have confirmed the mediating role of FoMO regarding the effect of social engagement on subjective well-being [19].

Based on the above discussion, we suggested that COVID-19 lockdown stress might impact college students' mental health through FoMO, and hypothesis 2 was proposed: COVID-19 lockdown stress was significantly positively correlated with FoMO; FoMO was significantly negatively correlated with mental health; and FoMO plays a mediating role between the COVID-19 lockdown stress and college students' mental health.

In conclusion, the following hypothetical model was constructed (Figure 1).

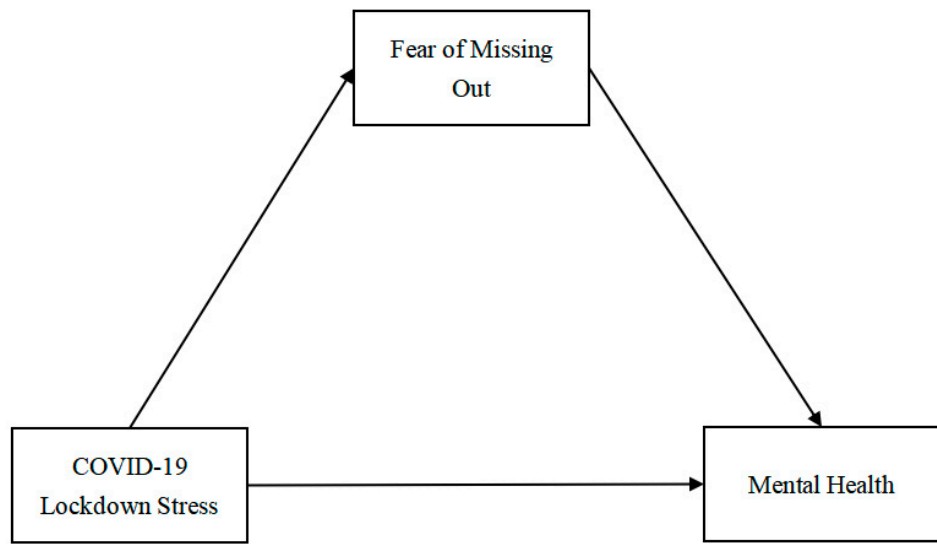

**Figure 1.** Hypothesis model.

### 3. Method

*3.1. Participants and Procedure*

We adopted the purposive sampling method to conduct a questionnaire survey among college students in Yunnan Province, China. The university is located in southwestern China, where cases of imported epidemics from outside the country continue to occur, and, therefore, the university is often under various degrees of containment. The data were collected from 10–12 May 2022, when the university was under lockdown, which restricted studying and social life.

This study was ratified and ethically reviewed by the research ethics committee of the affiliated department of the corresponding author (approval code: HKD-2022-25; approval date: 7 May 2022), following the fundamental principles of the Declaration of Helsinki [49]. First, we trained the teachers in charge of the survey before the questionnaire was distributed. The contents of training included the purpose of this study, the design of the questionnaire and the criteria for recruiting participants (college students who were interested in the research topic and volunteered). Next, the participants were told by their class teachers about the research purpose and confidentiality agreement before they filled out the questionnaire. They were also informed that questionnaire submission and data processing were conducted anonymously; they could refuse or withdraw at any point if they had any doubts when they were filling the questionnaires; and neither refusal nor withdrawal would have any adverse consequences. After the participants were informed and had consented, questionnaires were distributed to them via Questionnaire Star (www.wjx.cn, accessed on 31 July 2022).

A total of 760 questionnaires were administered, and the average completion time was 4 min. Completion times that were too short (<1 min and 30 s) or too long (>8 min), or incomplete questionnaires, were regarded as invalid. 65 invalid questionnaires were excluded, and 695 valid questionnaires were returned, achieving an effective rate of 91.45%. According to the formula proposed by Israel [50] for calculating the sample size, Yunnan Normal University has approximately 30,000 students (N ≈ 30,000), and the ideal sample size should be ≥652, so the valid sample size satisfied these criteria. Among the 695 valid responses, 167 (24.0%) and 528 (76.0%) respondents were male and female students, respectively; 232 (33.4%) were only-child status and 463 (66.6%) were non-only-child status; 167 (24%) were ethnic minorities and 528 (76%) were China's main nationality; 313 (45.0%) were first-year university students, 158 (22.7%) were second-year students, 151 (21.7%) were third-year students, and 30 (4.3%) were fourth-year students; and 43 were (6.2%) graduate students. The ages of all participants were between 18 and 27 years.

### 3.2. Instruments

### 3.2.1. COVID-19 Lockdown Stress

The 7-item COVID-19 student stress questionnaire (CSSQ) developed by Zurlo et al. (2022) was used to measure COVID-19 lockdown stress in Chinese college students. The questionnaire was designed to evaluate students' stressors of the COVID-19 lockdown with good reliability and validity [2]. There were three dimensions, namely, relationships and academic life, with four items; the isolation, with two items; and fear of contagion, with only one item. The questionnaire was scored on a 5-point Likert scale ranging from 1 (Not at all Stressful) to 5 (Extremely Stressful), with higher scores indicating more significant stress for college students due to the COVID-19 lockdown.

### 3.2.2. Mental Health

We employed the 6-item Mental Health Subscale of WHOQOL-BREF Quality of Life Assessment developed by WHOQOL Group in 1998 [51] to measure the mental health of Chinese college students. Lin [52] adapted it into a Chinese version of the 6-item mental health subscale and verified it with good reliability and validity among Chinese college students. The subscale was unidimensional, and the last question was reverse scored. It was a 5-point Likert scale, ranging from 1 (Strongly Disagree) to 5 (Strongly Agree), with higher scores indicating better mental health.

### 3.2.3. FoMO

The 10-item FoMO scale developed by Przybylski et al. [16] was used. The scale was unidimensional and was a 5-point Likert scale, ranging from 1 (Not at all True of Me) to 5 (Extremely True of Me), with higher scores indicating higher levels of FoMO.

### 3.3. Statistical Analysis

SPSS 21.0 (with statistical significance set at $p < 0.05$) was used for descriptive statistics, correlation analyses, Cronbach's $\alpha$ assess and the common method variance (CMV) test. Specifically, descriptive statistics were performed to reflect each variable's means and standard deviations; Pearson's correlation test was conducted to reflect the correlation between each variable and dimension. The correlation coefficient should be less than 0.7 and reach significance [53], indicating that all variables had a statistically significant correlation and no collinearity problem; Cronbach's $\alpha$ was greater than 0.7, indicating the excellent reliability of the measurement instrument [54]; Harman's one-factor test was conducted to assess common method variance (CMV), and the explanatory power of the first factor should not exceed the critical value of 50% [55].

In addition, this study used model 4 of Hayes' PROCESS, which is a plug-in of SPSS, to test the mediation model, with COVID-19 lockdown stress as the independent variable, FoMO as the mediating variable, and mental health as the dependent variable. The percentile bootstrap method at a bias-corrected confidence interval (CI) of 95% was

employed, and 2000 samples were used. The CI of each path coefficient should not contain 0, which indicated that the effect was significant [56].

AMOS 21.0 was used for the validity test (convergent validity and discriminant validity) and measurement model fitness evaluation. Specifically, convergent validity was needed to meet SRW (Standardized Regression Weight) > 0.5, CR (Composite Reliability) > 0.7, and AVE (Average Variance Extracted) > 0.5 [53]; according to the recommendation of Fornell & Laecker [57], the criterion of discriminant validity was: the square root of AVE for each dimension was larger than the correlation coefficient of each dimension, indicating that each scale or dimension had high discriminant validity. In addition, the value of $\chi^2$ was not reported in this study due to the sensitivity of the value of $\chi^2$ to a large sample size. Because the values of $\chi^2$ tend to reach significance when the sample size is large, other fitness indicators were verified [58]. According to Marsh et al. [59], the criteria for other fitness indicators were as follows: root mean square residual (RMR) < 0.08; comparative fit index (CFI) > 0.85; goodness-of-fit index (GFI) > 0.85; normed fit index (NFI) > 0.85; and incremental fit index (IFI) > 0.85. If the above criteria were satisfied, the fitness of the measurement model was acceptable.

According to Nitzl et al. [60] and Zhao et al. [61], mediation models were classified into different types as follows: (a) Full mediation; (b) Partial mediation (complementary partial mediation or competitive partial mediation)—complementary partial mediation was that the direct and indirect effects pointing in the same direction (same positive or negative), and competitive partial mediation was that the direct and indirect effects pointing in different directions; (c) Only direct effect; (d) No effect. This study also reported the type of the hypothesized mediation model.

## 4. Results

### 4.1. Measurement Model Assessment

4.1.1. COVID-19 Lockdown Stress

In the CFA, for the last item of the scale (How do you perceive the risk of contagion during this COVID-19 pandemic?), SRW (standardized regression weight) was <0.5 and was removed [62]. As shown in Table 1, for the other retained items, the values of SRW, CR (composite reliability), and AVE (average variance extracted) were 0.783–0.872 (>0.5), 0.888 and 0.859 (>0.7), and 0.665 and 0.753 (>0.5), respectively; according to the criteria of Cheung and Wang [53], the measurement model has high convergent validity. The fit indicators were RMR = 0.039, GFI = 0.976, CFI = 0.984, NFI = 0.981, and IFI = 0.984, indicating the appropriate fitness of the measurement model [59]. The value of Cronbach's $\alpha$ for the total scale in this study was 0.905 (>0.7), indicating good reliability [54].

**Table 1.** CFA of COVID-19 lockdown stress.

| Dimension | ITEM | SRW | CR | AVE |
|---|---|---|---|---|
| Relationships and Academic Life | How do you perceive the relationships with your university colleagues (peers) during this period of COVID-19 pandemic | 0.828 | 0.888 | 0.665 |
| | How do you perceive the relationships with your university professors (teachers) during this period of COVID-19 pandemic | 0.861 | | |
| | How do you perceive your academic studying experience during this period of COVID-19 pandemic | 0.788 | | |
| | How do you perceive the relationships with your relatives during this period of COVID-19 pandemic | 0.783 | | |
| Isolation | How do you perceive the changes in your sexual life due to the social isolation during this period of COVID-19 pandemic | 0.864 | 0.859 | 0.753 |
| | How do you perceive the condition of social isolation imposed during this period of COVID-19 pandemic | 0.872 | | |

Note: SRW = standardized regression weights; CR = composite reliability; AVE = average variance extracted.

### 4.1.2. Mental Health

The last item was reversed and removed because SRW was <0.5 [62]. The results of the CFA for the remaining items are shown in Table 2. The values of SRW were 0.722–0.885 (>0.5); CR = 0.902 (>0.7); and AVE = 0.649 (>0.5), indicating good convergent validity [53]. The fitness indicators of the measurement model were RMR = 0.061, GFI = 0.876, CFI = 0.892, NFI = 0.890, and IFI = 0.892, indicating an acceptable fitness [59]. The value of Cronbach's α for the total scale in this study was 0.903 (>0.7), indicating high reliability [54].

**Table 2.** CFA of mental health.

|  | ITEM | SRW | CR | AVE |
|---|---|---|---|---|
| Mental Health | I enjoy my life | 0.816 |  |  |
|  | I feel that my life is meaningful | 0.885 |  |  |
|  | I can concentrate (thinking, studying, remembering) on what I want to do | 0.851 | 0.902 | 0.649 |
|  | I can accept my appearance | 0.722 |  |  |
|  | I am satisfied with myself | 0.741 |  |  |

Note: SRW = standardized regression weights; CR = composite reliability; AVE = average variance extracted.

### 4.1.3. Fear of Missing Out

CFA results revealed that for two items of the original scale (questions 5 and 7), SRWs were <0.5, so these were deleted [62]. As shown in Table 3, for the retained items, the values of SRW were 0.582–0.850 (>0.5); CR = 0.903 (>0.7); and AVE = 0.545 (>0.5), indicating good convergent validity [53]. The fitness indicators were RMR = 0.073, GFI = 0.829, CFI = 0.855, NFI = 0.850, and IFI = 0.855, indicating an acceptable fitness [59]. The value of Cronbach's α for the total scale in this study was 0.905 (>0.7), indicating good reliability of the scale [54].

**Table 3.** CFA of FoMO.

|  | ITEM | SRW | CR | AVE |
|---|---|---|---|---|
| Fear of Missing Out | I fear others have more rewarding experiences than me | 0.828 |  |  |
|  | I fear my friends have more rewarding experiences than me | 0.846 |  |  |
|  | I get worried when I find out my friends are having fun without me | 0.822 |  |  |
|  | I get anxious when I don't know what my friends are up to | 0.850 |  |  |
|  | Sometimes, I wonder if I spend too much time keeping up with what is going on | 0.582 | 0.903 | 0.545 |
|  | When I have a good time it is important for me to share the details online (e.g., updating status) | 0.595 |  |  |
|  | When I miss out on a planned get-together it bothers me | 0.595 |  |  |
|  | When I go on vacation, I continue to keep tabs on what my friends are doing | 0.713 |  |  |

Note: SRW = standardized regression weights; CR = composite reliability; AVE = average variance extracted.

### 4.1.4. Discriminant Validity

The square root of AVE method was used to evaluate the discriminant validity. The square root of AVE for each dimension was larger than the correlation coefficient of each dimension (Table 4) [57], indicating that each scale or dimension had high discriminant validity.

**Table 4.** Discriminant validity.

| Dimension | *M* | *SD* | Relationships and Academic Life | Isolation | Mental Health | Fear of Missing Out |
|---|---|---|---|---|---|---|
| Relationships and Academic Life | 2.557 | 1.072 | *0.815* | | | |
| Isolation | 2.907 | 1.253 | 0.695 *** | *0.868* | | |
| Mental Health | 3.466 | 0.818 | −0.301 *** | −0.299 *** | *0.806* | |
| Fear of Missing Out | 2.695 | 0.796 | 0.440 *** | 0.382 *** | −0.208 *** | *0.738* |

Note: n = 695; the numbers in bold and italics in the diagonal are the square root of AVE. Numbers in the lower diagonal denote the correlation coefficients of two dimensions. *** $p < 0.001$; *M* = mean; *SD* = standard deviation.

### 4.2. CMV Test

To assess the CMV, Harman's one-factor test was used. Unrotated factor analysis indicated that the KMO (Kaiser–Meyer–Olkin) was 0.889 (>0.8); the Bartlett test of sphericity was significant ($p < 0.001$). The analysis received three factors of the characteristic root greater than 1, and the explanatory power of the first factor was 37.54%, which did not exceed 50% [55], indicating that no serious CMV problem existed.

### 4.3. Descriptive Statistics and Correlation Analysis

The descriptive statistics and correlation analysis of COVID-19 lockdown stress, mental health, FoMO, and gender are presented in Table 5. The results showed that COVID-19 lockdown stress and mental health were significantly and negatively correlated (r = −0.324, $p < 0.001$); COVID-19 lockdown stress and FoMO were significantly and positively correlated (r = 0.452, $p < 0.001$); and FoMO and mental health were significantly and negatively correlated (r = −0.208, $p < 0.001$). There was no significant correlation between gender and COVID-19 lockdown stress (r = 0.039, $p > 0.05$), mental health (r = −0.041, $p > 0.05$), or FoMO (r = −0.048, $p > 0.05$), so gender was not controlled for in the next step of regression analysis. The absolute values of the correlation coefficients were <0.7, indicating that there was no significant high correlation between the variables [53]. Therefore, there was no collinearity problem, satisfying the premise of conducting the regression analysis.

**Table 5.** Descriptive statistics and correlation analysis.

| Variable | *M* | *SD* | Gender | COVID-19 Lockdown Stress | Mental Health | Fear of Missing Out |
|---|---|---|---|---|---|---|
| Gender | 0.240 | 0.428 | 1 | | | |
| COVID-19 Lockdown Stress | 2.674 | 1.049 | 0.039 | 1 | | |
| Mental Health | 3.466 | 0.818 | −0.041 | −0.324 *** | 1 | |
| Fear of Missing Out | 2.695 | 0.796 | −0.048 | 0.452 *** | −0.208 *** | 1 |

Note: n = 695; gender was treated as a dummy variable, 1 = male, 0 = female; *** $p < 0.001$; *M* = mean; *SD* = standard deviation.

### 4.4. Mediating Role of FoMO

The mediating effect of FoMO was assessed using model 4 of the Hayes Process plug-in. The results are shown in Table 6 and Figure 2. In model 1, COVID-19 lockdown stress significantly and negatively predicted mental health (B = −0.256, $p < 0.001$); In model 2, COVID-19 lockdown stress significantly and positively predicted FoMO (B = 0.345, $p < 0.001$). When FoMO was added as a mediating variable in model 3, FoMO was a significant negative predictor of mental health (B = −0.104, $p < 0.05$), and COVID-19 lockdown stress was also a significant negative predictor of mental health (B = −0.220, $p < 0.001$), indicating that FoMO had a partial mediating effect between COVID-19 lockdown stress and mental health.

**Table 6.** Mediation model of FoMO.

| Variable | Model 1 Mental Health | | | Model 2 Fear of Missing Out | | | Model 3 Mental Health | | |
|---|---|---|---|---|---|---|---|---|---|
| | **B** | **SE** | **95% CI** | **B** | **SE** | **95% CI** | **B** | **SE** | **95% CI** |
| COVID-19 Lockdown Stress | −0.256 *** 0.028 (−0.328, −0.187) | | | 0.345 *** 0.026 (0.282, 0.410) | | | −0.220 *** 0.031 (−0.294, −0.145) | | |
| Fear of Missing Out | | | | | | | −0.104 * 0.040 (−0.210, −0.003) | | |
| $R^2$ | 0.108 | | | 0.200 | | | 0.116 | | |
| $f$ | 84.661 *** | | | 175.192 *** | | | 46.021 *** | | |

Note: B are unstandardized coefficients; SE, standard error; CI, confidence interval; * $p < 0.05$ *** $p < 0.001$.

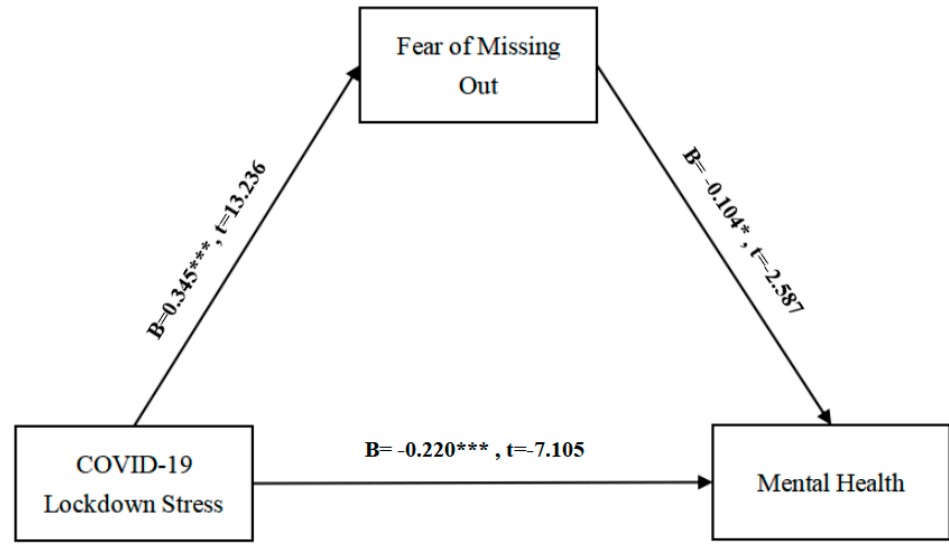

**Figure 2.** Mediation model of FoMO. * $p < 0.05$ *** $p < 0.001$.

The mediation effect of FoMO was tested again using the bootstrap method, and the results are shown in Table 7. The indirect effect value was −0.036, with 95% CI excluding 0 (−0.070, −0.005), indicating a mediating effect of FoMO. The direct effect value was −0.220, with 95% CI excluding 0 (−0.294, −0.145), verifying the partial mediating effect of FoMO. The total effect value was −0.256, with 95% CI excluding 0 (−0.328, −0.187), and the mediating effect accounted for 14.06%.

**Table 7.** Direct, indirect, and total effects.

| | **B** | **SE** | **95% CI** | |
|---|---|---|---|---|
| | | | **LL** | **UL** |
| Direct effect | −0.220 | 0.031 | −0.294 | −0.145 |
| Indirect effect | −0.036 | 0.020 | −0.070 | −0.005 |
| Total effect | −0.256 | 0.028 | −0.328 | −0.187 |

Note: B are unstandardized coefficients; SE, standard error; CI: confidence interval; LL: lower limit; UL: upper limit.

In addition, the direct effect (B = −0.220) and indirect effect (0.345 × −0.104 = −0.036) of COVID-19 lockdown stress on mental health were both negative (the direct and indirect effects pointed in the same direction), indicating that the mediation model of this study belongs to complementary partial mediation.

## 5. Discussion

The findings of this study support hypothesis 1 and suggest that COVID-19 lockdown stress has a significant negative impact on college students' mental health, similar to previous studies, reaffirming the significant predictive impact of stress on mental health in the context of major public health events (e.g., SARS) [4,39]. The possible reasons may be that (a) COVID-19 lockdown has led to dramatic changes in college students' learning and living environment [2]. Owing to all these sudden changes, college students' basic psychological needs, such as autonomy and social relationships, were not fully satisfied [2,32], which led to mental health problems [25]. (b) COVID-19 lockdown stress is enhanced in extreme environments [2,63,64], which may cause psychological discomfort and other health problems for college students.

The findings support hypothesis 2 and identify a complementary partial mediation model. COVID-19 lockdown stress not only directly and negatively affects college students' mental health, but also indirectly and negatively affects college students' mental health through FoMO. In previous studies, FoMO has been referred to as a state of unmet interpersonal needs [65], which is highly compatible with the isolated social context of COVID-19 lockdown. FoMO also increases the risk of psychological problems [38]. Specifically, policies such as closing all schools and restricting outdoor activities during the epidemic have restricted college students' normal social interaction with their peers [7], which will challenge their basic psychological needs (abilities, autonomy, relationships), and they may choose virtual social interaction as compensation [26]. However, the embellished image of others in virtual social networking may further highlight their own shortcomings, making them afraid of missing valuable experiences and triggering FoMO [17]. At the same time, FoMO will also promote the needs of interpersonal communication needs [47], which may lead college students into a vicious circle [26]. Therefore, COVID-19 lockdown stress may increase the degree of FoMO of college students and negatively affect their mental health.

The findings support and extend BPNT. First, we found a complementary mediating role of FoMO, suggesting that the two factors of FoMO and COVID-19 lockdown stress can comprehensively explain college students' mental health during the epidemic. Second, we regarded FoMO and the mental health problems of college students as psychological distress and discomfort caused by the unmet basic psychological needs caused by COVID-19 lockdown stress, confirming that environmental factors are crucial to college students' mental health [24,25,28].

## 6. Conclusions and Suggestions

This study explored the effects of COVID-19 lockdown stress on mental health and the mediating role of FoMO among Chinese college students. COVID-19 lockdown stress was shown to be significantly and negatively associated with mental health and significantly and positively associated with FoMO. Moreover, FoMO was significantly and negatively associated with mental health. COVID-19 lockdown stress directly and negatively predicted college students' mental health and also indirectly and negatively predicted college students' mental health by positively influencing FoMO. These findings support BPNT and provide empirical evidence for educators to develop effective preventive measures to address the mental health problems of college students during COVID-19 lockdown.

College teachers can strengthen policy interpretation during a particular period of a lockdown so that college students have a correct understanding of the lockdown and adequate psychological preparation. During the lockdown, students should be informed of practical difficulties that can be faced in learning or interpersonal communication, and effective academic guidance and emotional care should be offered.

Moreover, during the lockdown, collective activities could be widely carried out, such as online seminars or sharing activities on general knowledge of the COVID-19 epidemic to create opportunities for college students to communicate with each other and relieve their FoMO and further reduce unnecessary psychological discomfort.

Furthermore, colleges and universities should also focus on the autonomy of college students. Through the participation of student representatives in college management during the lockdown, the channels for students to make reasonable suggestions could be unblocked, reducing their sense of insecurity and isolation. At the same time, universities could set up special topics on their official websites to publicize the latest developments of the epidemic, guide college students to prevent the COVID-19 epidemic scientifically, and reduce unnecessary panic caused due to the pandemic.

## 7. Limitations and Future Research Directions

First, this study was only conducted among college students under lockdown in Yunnan Province, China, which implies some limitations in the generalization of the findings. Follow-up research may consider expanding the geographic scope of the sample (e.g., different regions in China or even other countries) or repeat the study in other groups (e.g., among high school students) to extend and validate the results of this study. Cross-cultural research may also be conducted to compare the differences under different culture and lockdown policy backgrounds in terms of the effects of COVID-19 lockdown stress on mental health.

Second, although the proportion of males to females in this study is in line with the biological gender composition of the sampling unit (the normal universities in China), in future research, universities with a balanced sample of male and female students should be selected as sampling institutions as far as possible. Gender balance will make research results more convincing.

Third, this study adopted a cross-sectional design, which meant that causal relationships between variables could not be derived. Longitudinal studies (e.g., cross-lag analysis) or experimental studies (e.g., analysis of covariance) could be considered for future research.

**Author Contributions:** Conceptualization, Z.H., J.L., L.P. and X.Z.; methodology, J.L.; formal analysis, Z.H. and J.L.; writing—original draft preparation, Z.H.; writing—review and editing, Z.H., J.L., L.P. and X.Z. All authors have read and agreed to the published version of the manuscript.

**Funding:** This research received no external funding.

**Institutional Review Board Statement:** This study was ratified and ethically reviewed by the research ethics committee of the affiliated department of the corresponding author (approval code: HKD-2022-25; approval date: 7 May 2022). Following the ethical standards required for conducting human research, this study followed the fundamental principles of the Declaration of Helsinki.

**Informed Consent Statement:** Informed consent was obtained from all subjects involved in the study.

**Data Availability Statement:** The data presented in this study are available on request from the corresponding author.

**Acknowledgments:** Thanks to all the participants in this study.

**Conflicts of Interest:** The authors declare no conflict of interest.

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
