# Peer review of "COVID-19 Lockdown Stress and the Mental Health of College Students: A Cross-Sectional Survey in China"

_sustainability, doi:10.3390/su141912923_

Round 1
Reviewer 1 Report
Please provide more demographic characteristics of the College students (specialization, year of study,….). Also, specify the number of students of the University (specify the name of the institution).
There is an official document (number, date) of the student management administration regarding the ethical agreement.
The discussion section could be enriched.
Author Response
请参阅附件

Reviewer 2 Report
In my opinion, the manuscript is interesting. The aim is to evaluate the relationship between the stress caused by isolation during the COVID-19 period and the mental health of Chinese students. The obtained results allow understanding the influence of the stress caused by isolation among Chinese students and the possibility of addressing this situation by university staff, with an emphasis on the basic psychological needs among students.
The manuscript with the title ,,COVID-19 Lockdown Stress and the Mental Health of College 2 Students: A Cross-Sectional Survey in China" is relatively well structured and analyzes the association between isolation stress due to COVID-19 and mental health among Chinese students, but requires minor modifications.
There are 61 references, of which 15 are more than 10 years old, 6 references between 5-10 years and 40 from the last 5 years. Does not include self-citation.
This study tracked the effects of isolation stress due to COVID-19 among Chinese students in the Yunnan region. Questionnaires were applied to a number of 695 students. Isolation stress was evaluated using a questionnaire developed by Zurlo and collaborators, identifying aspects related to relationships and academic life, isolation and fear of illness. The authors also used scales to assess the quality of life.
The statistical analysis was performed using SPSS 21.0. The obtained results were used to create a mediation model.
The results of the manuscript can represent a starting point for further research. The questionnaire method was used in the statistical analysis, the correlation, the variant test of the method, the Bootstrap method and the validity test were used.
The conclusions are consistent with those presented by the authors in the manuscript
The data in the tables are presented in detail and are easy to interpret.
However, there are several questions:
Lines 161-166: ,, First, we imparted training to the class teachers before recruitment. We explained the questionnaire’s items and recruitment criteria so that the class teachers could understand the purpose of the research and recruit participants who were interested in the research topic and would volunteer participation. Next, the participants were told by their class teachers about the research purpose and confidentiality agreement before they filled the questionnaire.
Can you elaborate ? wat do you mean when you say ,, we imparted training to the class teachers before recruitment. We explained the questionnaire’s items…
Lines 171-172: ,,A total of 760 questionnaires were administered, 65 invalid questionnaires were excluded, and 695 valid questionnaires were returned, ”.
Can you specify why they were invalid?
Lines 175-177 : ,, Among the 695 valid responses, 167 (24.0%) and 528 (76.0%) respondents were male and 175 female students, respectively; 232 (33.4%) were only children and 463 (66.6%) were non-only children; the ages of all participants were between 18 and 27 years”.
In the text you say that there were only 232 children and at the end of the paragraph you specify the age of 18-27 years. What age are you referring to when you talk about children? Did you refer to young people?
Lines 220-222 : ,, In the CFA, for the last item of the scale (How do you perceive the risk of contagion 220 during this COVID-19 pandemic?), SRW(standardized regression weight) was <0.5 and removed”.
You presented the results and specified the fact that you eliminated the last item of the scale for value <0.5.
Line 231: ,,The last item was reversed and removed because SRW was <0.5.”
And here you specify the fact that you eliminated the last item of the scale for value <0.5."
Lines 240-241: ,,CFA results revealed that for two items of the original scale (questions 5 and 7), SRWs 240 were <0.5, so these were deleted.”
The same specification as in the paragraphs above
However, in Table 4 Discriminant validity values <0.5 are also presented. Can you explain this aspect?
Lines 266-269: ,, There was no significant correlation between gender and COVID-19 lockdown stress, mental health, or FoMO, so gender was not controlled for in the next step of regression analysis. The absolute values of the correlation coefficients were <0.7, so there was no collinearity problem. “
Can you give more details? What did you think?
Lines 348-363: ,,First, this study was only conducted among college students under lockdown in Yunnan Province, China, which implies some limitations in the generalization of the findings. Follow-up research may consider expanding the geographic scope of the sample (e.g., different regions in China or even other countries) or repeat the study in other groups (e.g., among high school students) to extend and validate the results of this study. Cross-cultural research may also be conducted to compare the differences under different culture and lockdown policy backgrounds in terms of the effects of COVID-19 lockdown stress on mental health. Second, although the proportion of males to females in this study is in line with the biological gender composition of the sampling unit (the normal universities in China), distributing male and female college students as equally as possible in future studies would make the findings more representative. Third, this study adopted a cross-sectional design, which results that causal relationships between variables could not be derived. Longitudinal or experimental studies could be considered for future research”.
The limitations of the study and possibilities for research and continuation of the study are well specified. Do you want to continue the study as described in the text above?
Reviewer 3 Report
Thank you for your opportunity to review your interesting paper bringing together the concepts of mental health and fear of missing out among Chinese University students. An overall compelling paper, however, I have a few suggestions:
1) Introduction: Can you please reshape the introduction. In it's current state the introduction is a bit short and not yet specifically enough tailored towards the topic at hand. Can the authors please elaborate on the Covid-19 situation in China more precisely. Can the authors back with statistics on mental wellbeing of students in China? And how would the Covid-reality of students look like? And how has lockdown may have triggered fear of missing out? This is unclear to me? During lockdown there are no parties, social events, or leisure life with friends and family? What are the causes here to feel one misses out? Can the authors please give examples of causes in general and in Covidian times? If that is clearer maybe than the justification of your study becomes clearer to reader
2) Justification for the aim of your study- can you please elaborate in which context fear of missing out is well studied? And how this study makes a contribution to the body of literature more explicitly
3) A last note on the introduction, how does the paper relate to sustainability? The psychology aspect is obvious but the connection to sustainability or development is not explained. This connection is nowhere mentioned and justified. Can you please add the information?
4) Conceptual framework: Chapter 2.1: I understand that BPNT is a theoretical branch that frames this work? Why was it chosen? What were alternatives? Does a specific hypothesis is deducted from this theory? The last sentence is very general. Can this please rephrased into a more specific hypothesis if that was the intention?
5) Chapter 2.3: Maybe the following studies are helpful to further develop the chapter:
Hayran, C., & Anik, L. (2021). Well-being and fear of missing out (FOMO) on digital content in the time of COVID-19: A correlational analysis among university students. International journal of environmental research and public health, 18(4), 1974.
Gioia, F., Fioravanti, G., Casale, S., & Boursier, V. (2021). The effects of the fear of missing out on people's social networking sites use during the COVID-19 pandemic: the mediating role of online relational closeness and individuals' online communication attitude. Frontiers in Psychiatry, 12, 620442.
6) Method: Can you please explain more clearly who recruited the students? Did they receive incentives to participate? Class teachers have been mentioned involved? Isn't that ethical questionable due to power dynamics and dependencies?
7) Analysis: Can you elaborate on your analysis? Is the SEM or PLS-SEM? Can you please elaborate? And explain the analysis in more detail? You mention bootstrapping. How many iterations? If you diverge from standard procedure please explain. You mention you use fit indices. Please indicate which and justify.
8) Can the authors elaborate on further limitations of study and their sampling and analytical approach?
Round 2
Reviewer 3 Report
All my questions and concerns have been carefully addressed. I have no more reservations about the paper. An interesting piece of work!